# A Precision Surgery Framework for Lung Resection: Robotic, Video-Assisted, and Open Segmentectomy

**DOI:** 10.3390/jpm15080387

**Published:** 2025-08-19

**Authors:** Chiara Catelli, Miriana D’Alessandro, Federico Mathieu, Roberto Corzani, Marco Ghisalberti, Andrea Lloret Madrid, Susanna Guerrini, Piero Paladini, Luca Luzzi

**Affiliations:** 1Thoracic Surgery and Lung Transplant Unit, Department of Medicine, Surgery and Neurosciences, Azienda Ospedaliero-Universitaria Senese, University of Siena, 53100 Siena, Italy; f.mathieu@student.unisi.it (F.M.); marco.ghisalberti@ao-siena.toscana.it (M.G.); luca.luzzi@unisi.it (L.L.); 2Respiratory Diseases Unit, Department of Medicine, Surgery and Neurosciences, Azienda Ospedaliero-Universitaria Senese, University of Siena, 53100 Siena, Italy; mirian.dalessandro@unisi.it; 3Diagnostic Imaging Unit, Department of Medical Sciences, Azienda Ospedaliero-Universitaria Senese, University of Siena, 53100 Siena, Italy

**Keywords:** lung segmentectomy, VATS, RATS, lymph nodes, overall survival, minimally invasive surgery

## Abstract

**Objectives:** To evaluate outcomes of patients undergoing lung segmentectomy using open thoracotomy, Video-Assisted Thoracoscopic Surgery (VATS), or Robotic-Assisted Thoracoscopic Surgery (RATS) approaches. **Methods:** A total of 157 patients (mean age: 68.7 years; 58% male) who underwent lung segmentectomy from 2015 to 2024 at the Thoracic Surgery of Siena were retrospectively enrolled and divided into groups based on the surgical approach: thoracotomy (*n* = 60), VATS (*n* = 58), and RATS (*n* = 39). No significant differences were observed between groups in terms of age, gender, or tumor stage. Peri-operative outcomes, and, in patients with non-small cell lung cancer (NSCLC, *n* = 104), long-term outcomes, were analyzed. Group comparisons were conducted using Kruskal–Wallis, Dunn’s test, Chi-squared, or Fisher’s exact test and Kaplan–Meier analysis with log-rank test. **Results:** Conversion rate was 13% and 0% for VATS and RATS, respectively (*p* = 0.005). Pleural effusion on first post-operative day was lower in RATS than VATS (*p* = 0.0006) and open (*p* < 0.0001). The maximum Visual Analogue Scale (VAS) value recorded was lower in RATS than open (*p* = 0.016) and VATS (*p* = 0.013). Surgery time was longer for RATS than open (*p* = 0.001) and VATS (*p* = 0.013). No differences were found in hospital stay and post-operative complications. In patients with NSCLC, the median follow-up was 25 months. The 90-day mortality rate was 9.5% in thoracotomy, 0% in VATS and RATS (*p* = 0.05). The 1- and 2-year overall survival was higher in VATS and RATS groups than thoracotomy (*p* = 0.001 and *p* = 0.040, respectively). The number of harvested lymph nodes was larger in the open group (*p* = 0.010), while a higher number of stations were harvested in RATS and open than VATS (*p* = 0.001). No differences were found in local recurrence (*p*= 0.08). **Conclusions:** RATS segmentectomy ensures a lower conversion rate, less post-operative pain, reduced daily pleural effusion, and a greater number of harvested lymph node stations compared to VATS, providing comparable peri-operative outcomes. RATS and VATS segmentectomy offer an advantage over the open approach in short- and long-term survival.

## 1. Introduction

To date, several clinical trials, including the Japan Clinical Oncology Group (JCOG) series (JCOG0802/WJOG4607L and JCOG1211) and Cancer and Leukemia Group B (CALGB) 140503 [1,2], have demonstrated how segmentectomy has comparable oncological outcomes to lobectomy in the treatment of early-stage NSCLC, as long as sufficient surgical margins are ensured, and there are no lymph node metastases present. This procedure results in parenchymal sparing, thus preserving respiratory function. Segmentectomy is therefore particularly beneficial for patients with a limited cardiopulmonary reserve, such as those with chronic obstructive pulmonary disease, previous lung resections, or other comorbidities, where preserving as much functional lung tissue as possible is crucial to maintaining post-operative quality of life.

Segmentectomy is still a more challenging procedure due to the need to create one or multiple new intersegmental planes, which are often not easily identifiable [3]. Initially, many procedures were performed via open thoracotomy, as the VATS approach—despite being minimally invasive—was limited by two-dimensional vision, restricted instrument mobility, and reduced depth perception, which made complex segmental dissection and accurate lymphadenectomy more difficult. [4,5] The advent of Robotic-Assisted Thoracoscopic Surgery (RATS) as a minimally invasive technique has allowed for better 3D visualization, reduced intentional tremors, and ensured better lung manipulation. These features are particularly useful in segmentectomy, where the dissection of small vessels and bronchi, as well as the identification of intersegmental planes, is technically demanding. Enhanced 3D vision improves spatial orientation, while tremor reduction and wristed instruments enable more precise and controlled movements during complex anatomical maneuvers.

To date, there are few studies that compare the different surgical approaches (open, VATS, or RATS) for the execution of lobectomy and segmentectomy [6,7,8].

In the context of personalized medicine, the choice of surgical approach is becoming an increasingly crucial factor in optimizing outcomes for patients with NSCLC, especially in the presence of comorbidities or specific anatomical characteristics. Techniques such as RATS and VATS can be considered tools of surgical personalization, as they allow for tailoring the invasiveness of the procedure to individual clinical needs, maximizing functional preservation and minimizing operative risks. This study fits within the framework of personalized medicine by evaluating how different surgical approaches can be selected based on the clinical and anatomical specificities of each patient.

Our retrospective study aims to compare peri-operative and short-term outcomes of open, VATS, or RATS segmentectomy for primary or secondary lung tumors. The secondary endpoint is to evaluate oncologic outcomes in patients with NSCLC as a definitive pathology.

## 2. Materials and Methods

This retrospective study included all patients who underwent lung segmentectomy at the University Hospital of Siena from 1 January 2015 to 31 December 2023. Indications for pulmonary segmentectomy were based on clinical or oncological/radiological criteria per JCOG0802/WJOG4607L recommendations [1]. Patient operative tolerance was assessed via blood gas analysis, pulmonary function tests, electrocardiogram, and echocardiography. Segmentectomy was considered for patients with a forced expiratory volume in the first second (FEV_1_) or carbon monoxide diffusing capacity (DLCO) below 60% of the predicted value or, in cases where the suspected tumor was small, peripheral, and early stage, typically not exceeding 2 cm in diameter without evidence of lymph node involvement or metastasis.

All patients underwent standard pre-operative imaging, including contrast-enhanced chest CT scan and [18F]-FDG PET/CT for metabolic characterization of the lesion and evaluation of mediastinal and distant metastases. Invasive staging with mediastinoscopy or EBUS was reserved for cases with PET-positive or enlarged (>1 cm) lymph nodes on imaging. A brain CT or MRI was performed in patients with clinical suspicion of neurological involvement or stage ≥ IB disease.

Exclusion criteria included prior ipsilateral lung resection, concurrent lobectomy and segmentectomy, organizing pneumonia on intraoperative pathology, or prior neoadjuvant therapy. Patients were divided into three groups based on the surgical approach: open thoracotomy (Group 1), VATS (Group 2), and RATS (Group 3). Patients converted to open thoracotomy during surgery were considered within the open group in the outcomes’ analysis.

Given the retrospective, observational, and anonymized nature of the study, formal approval from the institutional ethics committee was not required, in accordance with current national regulations. All patients gave their informed consent for privacy and for the use of their clinical data for research purposes.

The type of surgical approach used (open, VATS, or RATS) was determined based on the first operator’s choice, reflecting their greater experience and familiarity with the selected technique. This approach allowed the lead surgeon to leverage their expertise to optimize procedural outcomes according to the specific characteristics of each case. All patients received general anesthesia with single-lung ventilation via a double-lumen endotracheal tube. Peri-operative analgesia was administered through epidural anesthesia for patients undergoing thoracotomy until 2018. After this period, the analgesic approach was modified to utilize intercostal block, applied not only to patients undergoing MIS but also to those receiving thoracotomy. The open thoracotomy involved an incision of about 10–12 cm (lateral or posterolateral, depending on the surgeon’s preference and the location of the lesion) performed on the upper margin of the sixth rib and spread with the use of retractors. The VATS segmentectomy procedures were carried out using the Copenhagen 3-port approach, with an anterior utility port of about 3–4 cm, spread with the use of a wound retractor. RATS segmentectomy was performed using four thoracoscopic accesses (three in the 8th intercostal space and one anteriorly in the 5th–6th space on the midaxillary line), without the use of CO_2_, using the da Vinci Xi Surgical System (DVSS) (Intuitive Surgical, Sunnyvale, CA, USA). Harmonic Ace™ by Ethicon (Raritan, NJ, USA) was used for dissection in VATS and RATS. The procedure involved segmentectomy followed by hilar and mediastinal lymphadenectomy for confirmed or suspected primary lung cancer.

All resections were anatomical segmentectomies, with broncho-vascular elements isolated and closed using mechanical suturing devices. The intersegmental plane was completed using parenchymal suturing devices after a lung re-ventilation test. In RATS, indocyanine green and Firefly mode were used for fluorescence to delineate intersegmental planes.

Peri-operative outcomes evaluated included

-Intraoperative outcomes: Conversion rate, type of segmentectomy (simple/complex), surgery duration, intraoperative complications.-Short-term outcomes: Pleural effusion, chest tube removal day, air leak duration, hemoglobin loss, hospital and ICU stay lengths, post-operative pain (VAS), post-operative complications.

A subgroup of patients with a definitive diagnosis of NSCLC was selected and divided by surgical approach for comparison of oncologic outcomes (90-day mortality, 1-year and 2-year OS, disease recurrence, oncologic stage, number of harvested lymph nodes/stations). All patients were staged per the 8th edition of the *AJCC Cancer Staging Manual*. All patients with a stage higher than IB were discussed within a multidisciplinary tumor board to assess their eligibility for adjuvant therapy based on clinical condition, comorbidities, and pathological findings. Patients with confirmed N2 disease were evaluated for post-operative radiotherapy on a case-by-case basis, in accordance with multidisciplinary tumor board decisions. Follow-up was conducted 1 month after surgery, biannually for 3 years, and annually thereafter. At each follow-up visit, a contrast-enhanced chest CT scan was performed every 6 months for the first 3 years and then annually. PET/CT was not routinely performed during follow-up but was reserved for cases with suspected recurrence based on clinical symptoms or CT findings. Brain imaging was only repeated if neurologic symptoms occurred. OS was calculated from the operation to death or last follow-up.

### Statistical Analysis

All data were expressed as mean ± standard deviation or median and interquartile range, when appropriate. A non-parametric one-way ANOVA (Kruskal–Wallis test) and Dunn test were performed for multiple comparisons. For categorical variables, we used Fisher’s exact or Chi-squared tests for intergroup comparisons. Spearman’s test was used to test the correlation of clinical and immunological parameters. Statistical analyses were performed using GraphPad Prism 10 software (GraphPad Software Inc., San Diego, CA, USA) and Jamovi 2.7.5 software. A *p* value ≤ 0.05 was considered statistically significant. Ninety-day mortality, as well as 1-year and 2-year survival rates, were calculated using Kaplan–Meier survival analysis and the log-rank test.

## 3. Results

### 3.1. Patients’ Characteristics

This study included 191 patients who underwent lung segmentectomy at University Hospital of Siena from 1 January 2015 to 31 December 2023. After excluding patients as outlined above, the remaining 157 patients were divided into three groups: open thoracotomy (*n* = 60), VATS (*n* = 58), and RATS (*n* = 39). Eight patients were converted to open thoracotomy during surgery. The characteristics of enrolled patients are summarized in Table 1.

No significant differences were found between the groups in terms of mean age (respectively, 68, 68, and 70 years for Groups 1, 2, and 3, *p* = 0.339). No differences were found in smoking habits, *p* = 0.964, and examined comorbidities (diabetes, atrial fibrillation, chronic bronchopulmonary disease, renal insufficiency, vasculopathy, history of previous neoplasia), except for the incidence of heart disease, which was higher in patients undergoing RATS and VATS compared to open surgery (respectively, 20.7% and 33.3% in VATS and RATS groups, 2.5% in open group, *p* = 0.003). Regarding the type of segmentectomy performed in the three groups, no significant differences were observed, with a more frequent execution of simple segmentectomy in each group (respectively, in 80%, 84%, and 82% of cases in Groups 1, 2, and 3, *p* = 0.817) compared to complex segmentectomy. In all three groups, a predominant diagnosis of lung adenocarcinoma was made, with 33 cases (55.0%) in Group 1, 34 cases in Group 2 (58.6%), 24 cases in Group 3 (61.5%), *p* = 0.0525, followed by squamous cell carcinoma, neuroendocrine tumor, and metastasis from solid neoplasm.

Figure 1 shows the annual number of segmentectomies performed, categorized by different surgical approaches.

### 3.2. Peri-Operative and Long-Term Outcomes

The peri-operative outcomes are summarized in Table 2.

The duration of the surgical procedure was found to be longer in RATS compared to VATS and open surgery, with an average of 153 min in open and VATS procedures and 189 min in RATS (*p* = 0.0005). Eighteen patients (32%) in the open group required ICU admission after surgery, while only seven (12%) and eight (20%) respectively in the VATS and RATS groups (*p* = 0.039) required this. ICU stay was shorter for patients undergoing VATS (mean 1.1 days) than open surgery (mean 2.5 days, *p* = 0.03), while no difference was found between VATS and RATS (mean 1.3 days, *p* = 0.446). In the first post-operative day, pleural fluid losses from the chest tube were significantly lower in the group of patients undergoing RATS compared to VATS and open (216 mL vs. 298 and 437 mL, respectively, *p* < 0.0001). On the second post-operative day, pleural fluid losses were also lower in the RATS group (*p* = 0.0039). However, pairwise comparison revealed that the difference was significant only compared to the open technique (202 mL and 267 mL for RATS and open, respectively, *p* = 0.002) but not to VATS (243 mL, *p* = 0.12). There were no significant differences in air leak days among the three groups, averaging less than one day in each group (*p* = 0.998), and in the duration of chest drainage, approximately four days in each group (*p* = 0.364).

No statistically significant differences were observed in the development of post-operative complications (*p* = 0.234): they occurred in three patients (5%) in Group 1 and consisted of prolonged air leaks, in one case complicated by the development of pneumonia. Post-operative complications were observed in eight patients (13.8%) in Group 2, including five cases of prolonged air leaks, two cases of pneumonia, and one case of post-operative atrial fibrillation. In Group 3, only three patients (7.7%) developed post-operative complications, including two prolonged air leaks and one atrial fibrillation. A blood patch procedure was executed in five patients (3.2%). There were no statistically significant differences in post-operative pain according to the VAS scale on the first post-operative day among the three groups (*p* = 0.149), nor at discharge (*p* = 0.398), while a significant difference emerged in the maximum pain perceived during hospitalization, with scores of 3.4 in the open and VATS groups and 2.6 in the RATS group (*p* = 0.007). The length of hospital stay was 6.1 days, 5.2 days, and 4.9 days for the open, VATS, and RATS groups, respectively (*p* = 0.487). Conversion to thoracotomy was necessary in eight cases (13%) in the VATS group and in no cases in the RATS group (*p* = 0.005). The difference in hemoglobin blood levels between admission and discharge was higher in the RATS group (−1.86 g/dL, *p* = 0.017); however, only one out of 157 patients (0.6%) required post-operative transfusions. When stratified by the type of segment performed (simple or complex), no significant difference was found in hemoglobin blood level (*p* = 0.30) or days of air leakage (*p* = 0.67), but there was a significant difference in the number of days of chest drainage, with a shorter duration for simple segments (*p* = 0.03). The 90-day mortality was lower in the RATS and VATS groups compared to the open group (0% and 6.7%, respectively, *p* = 0.03); the 1-year OS was 100% in the RATS group, 98.1% in the VATS group, and 88.0% in the open group (*p* = 0.023). There was no significant difference in 2-year OS between the three groups (80.9% in Group 1, 89.7% in Group 2, 100% in Group 3, *p* = 0.126).

### 3.3. Oncological Outcomes in Patients with NSCLC

The peri-operative and oncological outcomes for 104 patients with a definitive NSCLC diagnosis are summarized in Table 3.

Eleven patients (10.6%) underwent adjuvant treatments, nine in the open group and two in VATS group. Among them, two patients with pathologic N2 involvement also received post-operative radiotherapy (PORT). One elderly patient did not receive adjuvant chemotherapy due to age-related frailty and comorbidities. No significant differences were observed between stages (*p* = 0.095), though the RATS group showed a trend towards earlier-stage diagnoses. No differences were seen in pN stage (*p* = 0.397). More lymph nodes were harvested in the open group (mean 9.7 nodes, SD 6.2) compared to RATS (mean 6.6, SD 5.4) and VATS (mean 6.7, SD 4.5) (*p* = 0.0098), and more lymph node stations were removed in the open and RATS groups (mean 3.8, SD 1.3 stations and 3.4, SD 1.1, respectively) compared to VATS (2.8 stations, SD 1.1, *p* = 0.001).

In the first 90 days post-surgery, no deaths occurred in the VATS or RATS groups, while four deaths occurred in the open group (9.5%, *p* = 0.046). Significant differences were found in 1-year OS (84.6% open, 100% VATS, 100% RATS, *p* = 0.01) and 2-year OS (75.8% open, 96.2% VATS, 100% RATS, *p* = 0.04), with no deaths at 2 years in the RATS group. The number of harvested lymph nodes did not affect 1- and 2-year OS. Kaplan–Meier analysis showed no significant OS difference among the three techniques (*p* = 0.18) (Figure 2), but minimally invasive surgery (MIS) (VATS plus RATS) had a significant advantage over open surgery (*p* = 0.05) (Figure 3).

No statistically significant differences were observed among the three surgical groups in terms of disease recurrence (*p* = 0.086). However, the recurrence rate was notably lower in the robotic-assisted surgery (RATS) group (4%) compared to the VATS (24.3%) and open thoracotomy (23.8%) groups. The clinical and pathological characteristics of patients who experienced recurrence, including timing, histology, and surgical approach, are summarized in Table 4.

## 4. Discussion

Most studies conducted to date on segmentectomy only compare the VATS technique and thoracotomy, while only a few studies focus on the RATS approach. Our study analyzed patients undergoing pulmonary segmentectomy via open, VATS, and RATS techniques. The groups were similar in medical history, except for cardiac comorbidities: RATS patients had more heart disease compared to VATS and open. More open segmentectomy patients required ICU admission than those in the VATS or RATS groups, despite VATS and RATS patients being frailer because of cardiopathy. This suggests that MIS is less impactful on frail patients, confirming its advantage in terms of safety and tolerability, possibly due to reduced surgical invasiveness and lower cardiac trauma.

On the first post-operative day, pleural effusion was lower in RATS than in VATS and open surgeries, which might help maintain hydro-electrolytic balance and protect against post-operative arrhythmias [9]. However, RATS showed a greater reduction in pre- to post-operative hemoglobin levels compared to VATS and open (*p* = 0.017), indicating more blood loss, potentially due to a longer surgery duration (*p* = 0.0005) and different anesthetics management, after having excluded major intraoperative bleedings. Despite this, only one patient required a transfusion.

No differences were observed in chest tube duration, days of air leak, hospital stay length, or post-operative complications among the three groups, a result currently inconsistent with the literature [10,11,12]. This indicates that MIS techniques, despite being more complex than the open approach, do not lead to more intraoperative complications affecting the post-operative course.

The maximum pain reported on the VAS scale was statistically lower in the RATS group compared to VATS and open (*p* = 0.007). This is expected compared to open surgery, which requires rib spreading and muscle dissection, but less intuitive compared to VATS: in fact, RATS involves additional thoracoscopic access compared to the three-port VATS performed at our center. However, VATS involves a more frequent instrument introduction and removal than RATS, where robotic arms and trocars provide stability during instruments’ introduction, potentially reducing intercostal nerve irritation and post-operative pain.

Segmentectomy is more challenging than lobectomy due to smaller structures to dissect and less identifiable intersegmental planes. The robotic approach, with its three-dimensional vision, tremor elimination, and endo-wrist movement, could offer advantages in complex surgeries. Our study found a 13% conversion rate from VATS to open technique, compared to 0% for RATS, indicating greater safety and reliability for RATS. This advantage is independent of the type of segmentation performed. While this did not reduce hospital stays in our sample, it may do so in a larger population.

RATS procedures lasted longer than VATS or open (*p* = 0.0005), a result consistent with other studies [13]. This can be attributed to surgeons’ greater experience with open and VATS techniques and the limited number of RATS procedures executed (39, by three surgeons). Specifically, a total of 39 RATS segmentectomies were performed: 18 by Surgeon A, 10 by Surgeon B, and 11 by Surgeon C. All three surgeons had already completed their learning curve in robotic thoracic surgery—mainly through robotic lobectomy—but these cases represent the initial segmentectomies executed by each operator using the robotic approach. According to Zhang’s study [14], the learning curve for RATS segmentectomy requires 40 procedures. Our surgeons may still be within this learning curve, justifying the longer operative time, which could improve with experience [15].

Our study shows that MIS has a lower 90-day mortality rate compared to open surgery, with no deaths in the RATS and VATS. Additionally, 1-year OS was better in the RATS (100%) and VATS (98%) groups compared to the open group (88%, *p* = 0.023), like what was observed by Kent [16] in lobectomies. Although the latter advantage is lost at two years, it is interesting to note that no deaths were reported at two years in the RATS group.

To reduce the bias resulting from the different underlying pathologies regarding the extent of hilar and mediastinal lymphadenectomy and oncological outcomes, we selected a subgroup of patients (*n* = 104) undergoing segmentectomy for NSCLC. In this subgroup, it emerged that the number of lymph nodes retrieved during lymphadenectomy was higher in the open group compared to VATS and RATS (*p* = 0.0098). This result is consistent with the greater exposure of the surgical field in open surgery compared to VATS or RATS and also with the lower complexity of the surgical technique. However, it is important to consider that the data regarding the harvested number of lymph nodes is often imprecise, as individual lymph node fragments may be interpreted as single lymph nodes by the pathologist. If we instead consider the number of harvested lymph node stations, it was observed that RATS and open surgery have an advantage over VATS, a result that is consistent with that observed in Huang et al.’s meta-analysis [17]. This result highlights the significant advantage of RATS over VATS, namely the ability to reproduce an open procedure with a minimally invasive technique in terms of visibility and exposure of structures, factors that traditional VATS does not allow due to the necessary hand–eye coordination and the increased difficulty in vascular and lymph node dissection due to the absence of the endo-wrist movements of the robotic instruments [18]. RATS would therefore allow for optimal reproduction of lymph node dissection performed with an open approach, while combining all the advantages of a minimally invasive technique (less invasiveness, better tolerability, reduced post-operative pain). It is also possible that with the advancement along the learning curve, advantages in lymphadenectomy in terms of number of lymph nodes removed compared to VATS may also be achieved, as observed in several prospective trials [19]. Figure 1 highlights an increase in the total number of segmentectomies performed at our center during the period 2022–2023. Notably, there is a marked rise in RATS procedures in 2023, coinciding with a decrease in VATS procedures. Additionally, the graph shows an upward trend in open procedures in recent years compared to VATS, a reversal of the trend seen in prior years. This shift confirms a growing preference among surgeons for a reliable technique that allows for a comprehensive lymphadenectomy, which the literature identifies as a critical prognostic factor in lung cancer surgery.

Observing survival with a median follow-up of 25 months, in the NSCLC subgroup survival at 90 days (*p* = 0.0464), at one year (*p* = 0.0112), and at two years (*p* = 0.0403) is statistically superior in patients undergoing MIS compared to the open approach, with no deaths reported in the RATS group. If the difference is not significant when comparing the three surgical techniques individually (Figure 2), comparing MIS with open surgery, the Kaplan–Meier analysis demonstrates the significant advantage of the minimally invasive approach over the open approach (Figure 3).

This result highlights the advantage of MIS in both short-term and long-term survival for NSCLC patients. The number of lymph nodes removed did not affect OS, confirming that MIS, despite removing fewer lymph nodes, does not compromise oncological outcomes, staging, or resection radicality; this aligns with studies indicating that, particularly in segmentectomy, the total number of lymph nodes removed may not directly influence overall survival. For example, a recent analysis of thoracoscopic segmentectomy in early-stage NSCLC found that the number of dissected lymph nodes had no significant effect on overall or recurrence-free survival [20]. Additionally, Ding et al. reported that lymph node ratio—rather than the absolute number removed—was a significant prognostic factor in wedge resections but not in segmentectomy for tumors ≤ 2 cm [21]. These findings suggest that the adequacy of sampling and nodal evaluation, rather than sheer quantity, may be more determinant for oncologic outcomes in this setting.

The lower recurrence rate in the RATS group, although not significant, aligns with Cerfolio et al.’s study [22], which reported a 3.4% recurrence over a median 30-month follow-up. This suggests a potential advantage of RATS in ensuring adequate intersegmental plane closure to provide an adequate margin from the tumor, reducing the risk of local recurrence.

Analyzing the tumor staging distribution, the RATS group included more carefully selected patients, with no N^+^ pathology, unlike the VATS and open thoracotomy groups, which included five patients with advanced disease stages. Initially, segmentectomy was indicated for patients with compromised cardiac or respiratory function, often with neoplasms at a more advanced stage requiring an open approach to avoid compromising oncological radicality. However, recent studies [1,2] showed segmentectomy is non-inferior to lobectomy for early-stage NSCLC, leading to more patients with early-stage NSCLC undergoing MIS segmentectomy. This may bias the perceived advantage of MIS over open surgery in long-term OS, although not affecting short-term survival.

From a personalized medicine perspective, the ability to select the surgical approach based on the patient’s condition—such as functional reserve, cardiovascular comorbidities, or thoracic anatomy—represents a significant evolution from a standardized approach. In particular, our analysis shows that more fragile patients (e.g., those with heart disease) benefited from minimally invasive techniques, confirming the role of personalized surgery in clinical practice. The choice between RATS and VATS, in this context, is not solely based on technical considerations but reflects a true patient-centered treatment strategy.

### Study Limitations

This single-center, retrospective study had a limited sample size, precluding definitive conclusions. Propensity score matching was not performed, leading to potential bias from the sample size and patient characteristic variations. The analysis did not account for specific segmentectomy indications (oncological vs. comorbidities) nor differences in respiratory function, potentially biasing the survival analysis. A potential bias in this study is the inclusion of patients with varying cancer stages across different groups. This heterogeneity in oncological staging can influence outcomes, as prognosis and survival rates are often closely linked to disease stage at the time of treatment. Furthermore, while adjuvant treatment was administered to patients with stage IB–IIIB disease in accordance with national guidelines, one elderly patient was excluded due to comorbidities and age-related frailty. However, we did not have access to complete records detailing the specific chemotherapy or radiotherapy regimens used, as patients were referred to different oncologic units post-operatively. Similarly, molecular testing for oncogenic driver mutations in adenocarcinoma patients was performed selectively, depending on the year of diagnosis and disease stage, and data on the administration of targeted therapies are incomplete.

## 5. Conclusions

Robot-assisted segmentectomy demonstrated a lower conversion rate, reduced post-operative pain, decreased pleural effusion, and more comprehensive lymph node station dissection compared to VATS, while providing similar peri-operative and oncologic outcomes. Moreover, both RATS and VATS approaches were associated with improved short- and long-term survival compared to open thoracotomy. These findings underscore the growing role of minimally invasive surgery within the broader framework of personalized medicine. The ability to tailor the surgical approach—balancing invasiveness, precision, and oncologic radicality—based on patient-specific factors such as comorbidities, functional status, and anatomical characteristics, exemplifies a patient-centered strategy aimed at optimizing both clinical and quality-of-life outcomes. Robotic-Assisted Thoracic Surgery, in particular, combines the visual and technical advantages of open surgery with the benefits of a minimally invasive approach, offering an advanced tool for delivering personalized surgical care in early-stage NSCLC.

## Figures and Tables

**Figure 1 jpm-15-00387-f001:**
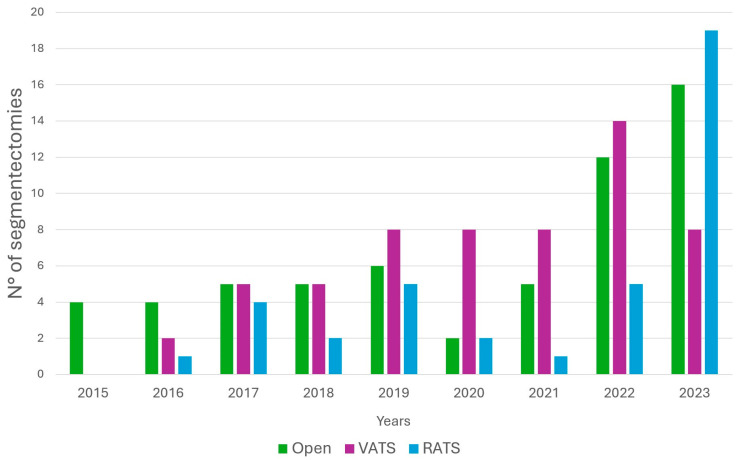
Lung segmentectomies performed per year, categorized by the surgical approach used (open thoracotomy, Video-Assisted Thoracoscopic Surgery, Robotic-Assisted Robotic Surgery).

**Figure 2 jpm-15-00387-f002:**
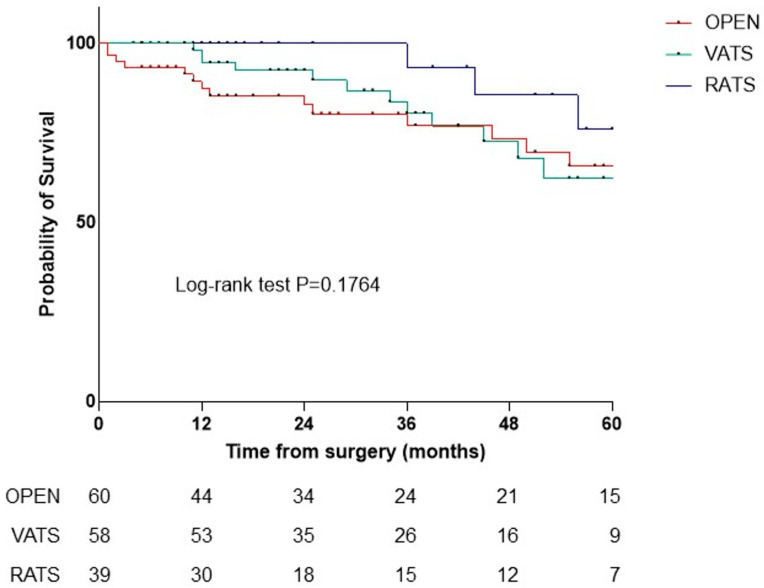
Overall survival of patients affected by NSCLC divided by surgical approach: RATS (blue), VATS (green), open (red).

**Figure 3 jpm-15-00387-f003:**
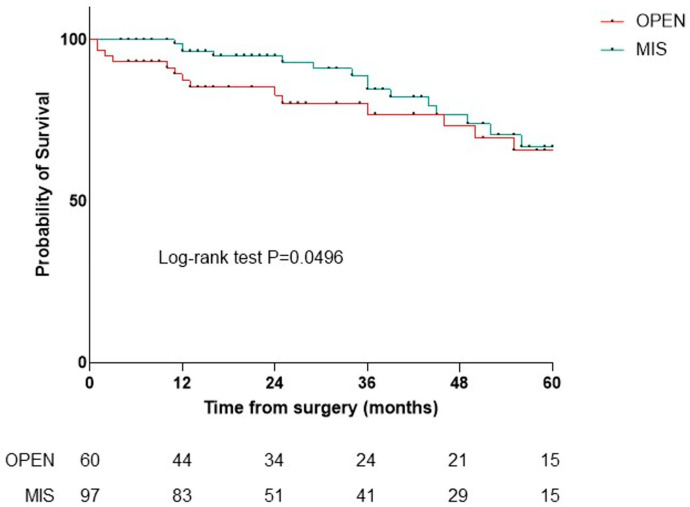
Overall survival of patients with NSCLC treated using open (red) or MIS (green) approaches.

**Table 1 jpm-15-00387-t001:** Patients’ characteristics.

Variables ^1^	Group 1 Thoracotomy (N = 60)	Group 2 VATS (N = 58)	Group 3 RATS (N = 39)	*p*
Age	68.7 (SD: 12.5)	68.7 (SD: 8.8)	70.1 (SD: 8.3)	0.399
Gender (m;f)	39;21	32;26	21;18	0.567
Smoke				0.964
Yes	35 (58.3%)	33 (56.9%)	20 (51.3%)	
Former	14 (23.3%)	13 (22.4%)	10 (25.6%)	
Cardiopathy	4 (2.5%)	12 (20.7%)	13 (33.3%)	0.003
Diabetes	6 (10.0%)	9 (15.5%)	7 (17.9%)	0.486
Atrial fibrillation	6 (10.0%)	2 (3.4%)	4 (10.2%)	0.466
Arterial hypertension	24 (40.0%)	34 (58.6%)	18 (46.2%)	0.123
COPD	11 (18.3%)	20 (34.5%)	11 (28.2%)	0.137
Renal failure	2 (3.3%)	1 (1.7%)	2 (5.1%)	0.643
Vasculopathy	22 (36.7%)	11 (19.0%)	13 (33.3%)	0.087
History of malignancy	21 (35.0%)	25 (43.1%)	13 (33.3%)	0.542
Type of segmentectomy (%)				0.817
Simple	48 (80%)	49 (84%)	32 (82%)	
Complex	12 (20%)	9 (16%)	7 (18%)	
Histopathology				0.053
Invasive adenocarcinoma	33 (55.0%)	34 (58.6%)	24 (61.5%)	
Squamous cell carcinoma	9 (15.0%)	3 (5.2%)	1 (2.6%)	
Neuroendocrine tumor	3 (5.0%)	3 (5.2%)	4 (10.3%)	
Metastasis	6 (10.0%)	13 (22.4%)	2 (5.1%)	
Other ^a^	9 (15.0%)	5 (8.6%)	8 (20.5%)	

^1^ The data are shown as absolute numbers (with percentage) for categorical variables and as mean ± standard deviation (SD) for continuous variables. Other ^a^: Adenoma, atypical adenomatous hyperplasia.

**Table 2 jpm-15-00387-t002:** Surgical and long-term outcomes.

Variables ^1^	Group 1 Thoracotomy (N = 60)	Group 2 VATS (N = 58)	Group 3 RATS (N = 39)	*p*
Surgery time (min)	153 (39)	153 (36)	189 (52)	0.001
ICU	18 (32%)	7 (12%)	8 (20%)	0.039
ICU days	2.5 (0.6)	1.1 (0.4)	1.3 (0.6)	0.032
Pleural fluid 1st pod (mL)	437 (203)	298 (133)	216 (159)	<0.0001
Pleural fluid 2nd pod (mL)	267 (106)	243 (130)	202 (145)	0.004
Air loss (days)	0.6 (1.3)	0.6 (1.4)	0.4 (0.7)	0.998
Length of chest tube (days)	4.2 (1.9)	4.1 (0.7)	3.9 (1.8)	0.364
Post-operative complications	3 (5.0%)	8 (13.8%)	3 (7.7%)	0.234
Blood patch	2 (3.3%)	1 (1.7%)	2 (5.1%)	0.643
VAS 1st pod	2.2 (1.4)	2.3 (1.3)	1.8 (0.9)	0.149
VAS max	3.4 (1.4)	3.4 (1.7)	2.6 (1.1)	0.007
VAS at discharge	1.0 (1.1)	0.8 (1.1)	0.8 (1.4)	0.398
LOS (days)	6.1 (5.2)	5.2 (2.2)	4.9 (1.9)	0.497
Conversion to open thoracotomy	-	8 (13%)	0 (0%)	0.005
Delta pre-operative and discharge hemoglobin (g/dL)	−1.36 (1.04)	−1.48 (0.76)	−1.86 (0.71)	0.017
90-days mortality	6.7%	0%	0%	0.036
1-year OS	88%	98.1%	100%	0.023
2-year OS	80.9%	89.7%	100%	0.126

^1^ The data are shown as absolute numbers (with percentage) for categorical variables and as mean with standard deviation for continuous variables. ICU: Intensive Care Unit; OS: overall survival; VAS: Visual Analogue Scale.

**Table 3 jpm-15-00387-t003:** Surgical and oncological outcomes in patients with NSCLC.

Variables ^1^	Group 1 Thoracotomy (N = 42)	Group 2 VATS (N = 37)	Group 3 RATS (N = 25)	*p*
Pathological stage				0.095
IA1	8 (19.0%)	12 (32.4%)	12 (48%)	
IA2	12 (28.6%)	14 (37.8%)	6 (24%)	
IA3	1 (2.4%)	2 (5.4%)	3 (12%)	
IB	7 (16.7%)	6 (16.2%)	3 (12%)	
IIA	4 (9.5%)	0 (0%)	1 (4%)	
IIB	5 (11.9%)	1 (2.7%)	0 (0%)	
IIIA	4 (9.5%)	1 (2.7%)	0 (0%)	
IIIB	0 (0%)	1 (2.7%)	0 (0%)	
N-stage				0.397
0	38 (90.4%)	34 (91.9%)	25 (100%)	
1	3 (7.2%)	1 (2.7%)	0 (0%)	
2	1 (2.4%)	2 (5.4%)	0 (0%)	
Number of harvested lymph nodes (N1, N2)	9.7 (SD: 6.2)	6.6 (SD: 5.4)	6.7 (SD: 4.5)	0.010
Number of lymph nodal stations harvested	3.8 (SD: 1.3)	2.8 (SD: 1.1)	3.4 (SD: 1.1)	0.001
90-days mortality	9.5%	0%	0%	0.046
1-year OS	84.8%	100%	100%	0.011
2-year OS	75.8%	96.2%	100%	0.040
Recurrence	10 (23.8%)	9 (24.3%)	1 (4%)	0.086

^1^ The data are shown as absolute numbers (with percentage) for categorical variables and as mean with standard deviation for continuous variables. OS: Overall Survival.

**Table 4 jpm-15-00387-t004:** Tumor and recurrence characteristics of patients with disease relapse.

Variables	Total Recurrences N = 20
Recurrence pattern	
Local: parenchyma	10 (50%)
Local: lymph node	5 (25%)
At distance	5 (25%)
Histopathology	
Adenocarcinoma	17 (85%)
Squamous cell carcinoma	3 (15%)
Staging	
IA	11 (55%)
IB	2 (10%)
IIB	4 (20%)
IIIA	3 (15%)
Previous adjuvant treatment	
Yes	7 (35%)
No	13 (65%)
Data is shown as number with percentages.	

## Data Availability

The original contributions presented in this study are included in the article. Further inquiries can be directed to the corresponding authors.

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
