# Peer review of "A Precision Surgery Framework for Lung Resection: Robotic, Video-Assisted, and Open Segmentectomy"

_jpm, 2025, doi:10.3390/jpm15080387_

Round 1

Reviewer 1 Report

Comments and Suggestions for Authors

I was glad to review this retrospective study which aims to evaluate outcomes of patients undergoing lung segmentectomy using open thoracotomy, Video-Assisted Thoracoscopic Surgery (VATS), or Robotic-Assisted Thoracoscopic Surgery (RATS) approaches. The manuscript is well-writen.
1) I would suggest in the abstract section to add patient characteristics, such as age, gender, or tumor stage in order to be able to compare the three surgical groups.
2)While p-values are provided, the abstract doesn't state what statistical tests were used to compare the groups in order to assess the validity of the results.
3)I would suggest adding a brief discussion about the rationale for choosing segmentectomy, specifically by elaborating on the patient populations for whom preserving respiratory function is a critical consideration.
4)I would suggest adding a more robust discussion on how the specific advantages of RATS, such as enhanced 3D visualization and tremor reduction, are hypothesized to directly overcome the technical challenges of segmentectomy in the introduction section.
5)I would suggest strengthening the introduction by adding citations to support the claims regarding the technical challenges of segmentectomy and the difficulties associated with the VATS approach.
6)Minor spelling errors like in line 21, 275, 298, 281.
7)I would suggest including specific details on the number of RATS procedures the three surgeons had performed both before and during the study period in the discussion section.
8)I would suggest expanding on the key paradox that open surgery harvested more lymph nodes but was associated with worse survival outcomes,  by providing a more detailed explanation or a broader literature review that supports the claim that the number of lymph nodes removed did not affect overall survival.

Author Response

Comment 1: I would suggest in the abstract section to add patient characteristics, such as age, gender, or tumor stage in order to be able to compare the three surgical groups.

Answer 1:  We have revised the abstract to include key patient demographics (mean age, gender distribution) and confirmed that no significant differences were found among the three surgical groups in terms of age, gender, and tumor stage. This information allows for better contextualization and comparability of the cohorts. (Lines 18-19; 21-22)

Comment 2: While p-values are provided, the abstract doesn't state what statistical tests were used to compare the groups in order to assess the validity of the results

Answer 2: The abstract has been updated to specify the statistical methods used for intergroup comparisons. These include the Kruskal-Wallis test, Dunn’s post-hoc test, Chi-squared or Fisher’s exact tests for categorical variables, and Kaplan-Meier analysis with log-rank test for survival outcomes. (lines 22-26).

Comment 3: I would suggest adding a brief discussion about the rationale for choosing segmentectomy, specifically by elaborating on the patient populations for whom preserving respiratory function is a critical consideration.

Answer 3: We appreciate the Reviewer’s valuable comment. In response to the suggestion, we have added a brief discussion in the Introduction to clarify the rationale for selecting segmentectomy, particularly in patients where preserving respiratory function is a critical consideration (lines 52-56)

Comment 4: I would suggest adding a more robust discussion on how the specific advantages of RATS, such as enhanced 3D visualization and tremor reduction, are hypothesized to directly overcome the technical challenges of segmentectomy in the introduction section.

Answer 4: We thank the Reviewer for the constructive suggestion. In response, we have expanded the Introduction to more clearly discuss how the technical advantages of RATS help overcome the inherent challenges of segmentectomy (lines 64-68)

Comment 5: I would suggest strengthening the introduction by adding citations to support the claims regarding the technical challenges of segmentectomy and the difficulties associated with the VATS approach.

Answer 5: We thank the Reviewer for this helpful suggestion. In response, we have slightly revised the Introduction to better support the statements regarding the technical challenges of segmentectomy and the limitations of the VATS approach. Specifically, we incorporated two additional references to substantiate these points (references 4 and 5).

Comment 6: Minor spelling errors like in line 21, 275, 298, 281

Answer 6: Thank you, we have corrected the spelling errors.

Comment 7: I would suggest including specific details on the number of RATS procedures the three surgeons had performed both before and during the study period in the discussion section.

Answer 7: We thank the Reviewer for this thoughtful comment. In response, we have included specific details in the Discussion section regarding the surgical experience of the three operators who performed RATS procedures.

Comment 8: I would suggest expanding on the key paradox that open surgery harvested more lymph nodes but was associated with worse survival outcomes,  by providing a more detailed explanation or a broader literature review that supports the claim that the number of lymph nodes removed did not affect overall survival.

Answer 8: We appreciate the Reviewer’s insightful comment regarding the apparent paradox whereby open surgery yielded more harvested lymph nodes yet was associated with worse survival outcomes, and their suggestion to deepen the discussion by referencing literature showing that the number of lymph nodes removed does not necessarily impact overall survival. In response, we have expanded the Discussion section to provide a more nuanced explanation supported by current evidence (references 20 and 21)

Reviewer 2 Report

Comments and Suggestions for Authors

The authors presented very interesting results for different surgical technics especially for patients with comorbidities. In the past majority of them were not candidates for surgery due to fragility. Thus this results are very important considering that lung cancer is the deadliest among cancers.

However I have some suggestion:

Please add in the section materials and methods what imaging methods were used before surgery (CT scan of the chest? PET/CT?, CT scan/MRI of the brain?). How you determine the lymph node positivity? Was mediastinoscopy performed?

How you followed the patients? How often you performed CT scan? Did you perform PET/CT in the follow up?

Did patients with stage Ib-IIIb received adjuvant treatment? If yes which therapy was used? In patients with adenocarcinoma did you perform molecular testing and if yes did patients with positive findings receive targeted therapy? Did patients with N2 status receive PORT?

What was the recurrence pattern? 

Please make other table for 1- and 2-year OS, for adjuvant treatment used and recurrence pattern. Also add information about the patients who had recurrence of disease (operative stage, adjuvant treatment, histopathology, recurrence pattern).

Author Response

Comment 1: Please add in the section materials and methods what imaging methods were used before surgery (CT scan of the chest? PET/CT?, CT scan/MRI of the brain?). How you determine the lymph node positivity? Was mediastinoscopy performed?

Answer 1: We thank the Reviewer for this important observation. In response, we have updated the Materials and Methods section to include detailed information on the preoperative imaging and mediastinal staging protocols. (lines 97-102)

Comment 2: How you followed the patients? How often you performed CT scan? Did you perform PET/CT in the follow up?

Answer 2: We thank the Reviewer for the pertinent comment regarding the follow-up protocol. In response, we have specified in the Materials and Methods section that patients were followed with contrast-enhanced chest CT every 6 months for the first 3 years, and annually thereafter. PET/CT was not routinely used during follow-up, but was selectively performed in cases with clinical or radiological suspicion of recurrence. Brain imaging was repeated only if neurologic symptoms developed. (lines 153-157)

Comment 3: Did patients with stage Ib-IIIb received adjuvant treatment? If yes which therapy was used? In patients with adenocarcinoma did you perform molecular testing and if yes did patients with positive findings receive targeted therapy? Did patients with N2 status receive PORT?

Answer 3 

  • Adjuvant treatment for stage IB–IIIB: All patients with stage higher than IB were discussed within a multidisciplinary tumor board to assess their eligibility for adjuvant therapy based on clinical condition, comorbidities, and pathological findings. (lines 148-150) Unfortunately, detailed records on the specific chemotherapy regimens administered to each patient are not fully available across the study cohort, as patients were referred to different oncologic centers for postoperative treatment. We have inserted a statement in the limitation of the study. (lines 430-438)

  • Molecular testing in adenocarcinoma: During the study period, molecular testing (including EGFR, ALK, ROS1, and PD-L1 status) was performed selectively in patients with adenocarcinoma, especially in advanced stages or in case of recurrence. However, due to the retrospective nature of the study and the evolving standard of care over time, not all patients underwent uniform molecular profiling. As a result, we do not have complete data on the use of targeted therapies in patients with actionable mutations.

  • Postoperative radiotherapy in N2 disease: Patients with confirmed N2 disease were evaluated for postoperative radiotherapy on a case-by-case basis, in accordance with multidisciplinary tumor board decisions. (lines 150-152) Some patients received PORT (2 patients), for extracapsular nodal extension, but detailed information regarding timing, dose, and protocol is not consistently available.

Comment 4: What was the recurrence pattern? Please make other table for 1- and 2-year OS, for adjuvant treatment used and recurrence pattern. Also add information about the patients who had recurrence of disease (operative stage, adjuvant treatment, histopathology, recurrence pattern).

Answer 4: We thank the Reviewer for this constructive suggestion. In response, we have added a new table (Table 4) that provides a detailed overview of the patients who experienced disease recurrence. The table includes data on operative stage, adjuvant treatment, histopathology, and recurrence pattern.

Round 2

Reviewer 2 Report

Comments and Suggestions for Authors

No further comments